# Relating Suprathreshold Auditory Processing Abilities to Speech Understanding in Competition

**DOI:** 10.3390/brainsci12060695

**Published:** 2022-05-27

**Authors:** Frederick J. Gallun, Laura Coco, Tess K. Koerner, E. Sebastian Lelo de Larrea-Mancera, Michelle R. Molis, David A. Eddins, Aaron R. Seitz

**Affiliations:** 1Oregon Hearing Research Center, Oregon Health & Science University, Portland, OR 97239, USA; coco@ohsu.edu (L.C.); koernert@ohsu.edu (T.K.K.); 2VA RR&D National Center for Rehabilitative Auditory Research, VA Portland Health Care System, Portland, OR 97239, USA; michelle.molis@va.gov; 3Department of Psychology, University of California, Riverside, CA 92521, USA; elelo001@ucr.edu (E.S.L.d.L.-M.); aseitz@ucr.edu (A.R.S.); 4Department of Communication Science & Disorders, University of South Florida, Tampa, FL 33620, USA; deddins@usf.edu

**Keywords:** auditory processing, hearing loss, speech perception, aging

## Abstract

(1) Background: Difficulty hearing in noise is exacerbated in older adults. Older adults are more likely to have audiometric hearing loss, although some individuals with normal pure-tone audiograms also have difficulty perceiving speech in noise. Additional variables also likely account for speech understanding in noise. It has been suggested that one important class of variables is the ability to process auditory information once it has been detected. Here, we tested a set of these “suprathreshold” auditory processing abilities and related them to performance on a two-part test of speech understanding in competition with and without spatial separation of the target and masking speech. Testing was administered in the Portable Automated Rapid Testing (PART) application developed by our team; PART facilitates psychoacoustic assessments of auditory processing. (2) Methods: Forty-one individuals (average age 51 years), completed assessments of sensitivity to temporal fine structure (TFS) and spectrotemporal modulation (STM) detection via an iPad running the PART application. Statistical models were used to evaluate the strength of associations between performance on the auditory processing tasks and speech understanding in competition. Age and pure-tone-average (PTA) were also included as potential predictors. (3) Results: The model providing the best fit also included age and a measure of diotic frequency modulation (FM) detection but none of the other potential predictors. However, even the best fitting models accounted for 31% or less of the variance, supporting work suggesting that other variables (e.g., cognitive processing abilities) also contribute significantly to speech understanding in noise. (4) Conclusions: The results of the current study do not provide strong support for previous suggestions that suprathreshold processing abilities alone can be used to explain difficulties in speech understanding in competition among older adults. This discrepancy could be due to the speech tests used, the listeners tested, or the suprathreshold tests chosen. Future work with larger numbers of participants is warranted, including a range of cognitive tests and additional assessments of suprathreshold auditory processing abilities.

## 1. Introduction

The ability to understand speech in the presence of competing sounds is a fundamental aspect of hearing that many older adults find challenging. Both age and degree of hearing loss have been shown to reliably predict a portion of the variability in performance on tasks of speech perception and have been studied extensively (for a review of earlier work, see Gordon-Salant [1]; for a more recent example, see Goossens et al. [2]). Hearing loss—customarily quantified on an audiogram as increased pure-tone detection thresholds—impacts speech perception through decreased overall audibility and reduced access to the temporal and spectral cues in speech. Similarly, normal aging can reduce the ability to use the cues needed for speech understanding in competition, although not all older listeners experience difficultly understanding speech in noise. 

Previous experiments conducted in our laboratories and elsewhere have identified a set of psychophysical tasks of auditory processing (so-called “suprathreshold” auditory processing tests) that appear to predict performance on speech tasks with either noise or speech as the interference [2,3,4,5,6,7]. However, with a few notable exceptions—Rönnberg et al. and Marsja et al. [8,9]—most of these investigations have either explored no more than one or two auditory processing tasks at a time, or have lacked the statistical power to allow firm conclusions to be drawn (for an example, see Neher et al. [7]) making it difficult to determine which independent predictors contribute most to speech-in-competition performance. To address the limitations of traditional laboratory-based psychophysical assessments of auditory processing ability, we developed a new assessment platform—Portable Automated Rapid Testing (PART)—that allows direct testing of a diverse range of auditory processing abilities and facilitates the collection of large data sets with consistent procedures. 

PART is an example of recent efforts to harness accessible technologies for hearing assessment in basic science and clinical telemedicine contexts, a current summary of which was recently created by the Acoustical Society of America’s Task Force on Remote Testing [10]. The PART application has already been used successfully to collect data both in the laboratory [11,12,13,14] and remotely with participant-owned equipment in their own homes [15,16]. Currently in the United States, the limited availability of hearing healthcare professionals relative to the number of people with hearing loss limits access to hearing healthcare, particularly in rural areas—a gap that will be exacerbated by the projected growth of the aging population [17,18,19]. The use of portable technologies such as PART can improve access to hearing healthcare for patients who have difficulty traveling to the clinic and will allow research investigators to recruit and test a more diverse population of study participants [18,19].

### Evidence for the Influence of Suprathreshold Auditory Abilities on Speech in Competition Performance

Behavioral and neurophysiological studies in humans and non-human primates have revealed substantial changes in brain structure, neurochemistry, and function associated with normal aging, even in the presence of normal or near-normal sensitivity or detection thresholds to pure tones (audibility) [20,21,22,23,24,25,26,27,28,29]. An additional set of auditory abilities is needed to discriminate different aspects of an audible signal. Since these processes operate on sounds that are above the audibility thresholds, they are referred to as suprathreshold auditory abilities. They include modulations of the amplitude of the acoustic signal over time (temporal modulation; TM) and modulation of the frequency spectrum of the acoustic signal relative to an unmodulated reference signal (spectral modulation; SM). Modulation of the spectrum that changes over time is called spectrotemporal modulation (STM). In addition, it is possible to modulate the phase of the signal, producing modulation in frequency of a pure tone or a narrowband noise (frequency modulation; FM). When the phase modulation is applied to both ears simultaneously it is called ‘diotic’, while applying FM to the signal at one ear, or different FM to the two ears is called ‘dichotic’. See Palandrani et al. [30] for more details on these types of signals. Another way of thinking about the modulation of the signal is through what has been called modulation of the temporal fine structure (TFS). See Hoover et al. [31] for a discussion and investigation of a wide range of tests of TFS sensitivity.

It is not yet clear how speech understanding depends on the ability to detect and discriminate these types of acoustic signals. However, speech signals contain all of these types of modulations, and neurophysiological evidence shows that the auditory system is very sensitive to all of these modulations. For example, STM provides a robust signal for the characterization of auditory cortical receptive fields [32,33,34,35,36,37]. STM-based representations of acoustical stimuli have also successfully been applied to computational models of speech representation [38,39]. Furthermore, the ability to detect STM has been shown to be related to speech understanding in listeners with normal pure-tone detection thresholds [40,41,42], in listeners with cochlear damage [3,4,43], and in listeners who use cochlear implants [44,45,46]. 

Sensitivity to TFS also is related to speech understanding, especially in competition [47,48,49]. TFS sensitivity relies on precise neuronal firing in the time domain—the most precise of all neurosensory systems in healthy listeners [50]. Neurophysiological studies in animals and electrophysiological studies in humans show that aging is associated with degradation in precise temporal coding [21,51,52,53]. Introducing TFS distortion also disrupts speech understanding, which has been interpreted as evidence that aging leads to reduced neural synchrony and thus increased temporal jitter [54,55]. Füllgrabe et al. [6] related the perception of TFS cues to speech understanding in the presence of masking. Sensitivity to differences in time of arrival at the two ears (“interaural timing differences”; ITDs) is based on the same cues as is dichotic FM detection, which has been shown to be a reliable method of measuring TFS sensitivity [31]. Ellinger et al. [56] showed that when only presented with ITDs, older listeners experienced less spatial release from masking than did younger listeners while Eddins and Eddins [57] showed that when TFS and envelope modulation cues convey binaural information simultaneously, coding of TFS cues accounts for age-related deficits in binaural release from masking. These studies support a role for TFS cues in speech understanding, especially in the presence of competing sounds.

The goal of this study was to better understand the relationships among these suprathreshold auditory processing abilities and speech understanding in the presence of competing speech. To do so, we explored the relationships between a test of speech understanding in competition (with and without spatial separation among the talkers) and a battery of six suprathreshold auditory processing tests, all implemented in the PART application. The six tests in the battery were selected based on prior studies demonstrating their potential to predict the outcome of measures of speech understanding in competition [1,19,20]. SM, TM, and STM detection was measured using tasks similar to those in the literature [3,58,59,60,61,62,63,64]. TFS sensitivity was measured using diotic FM as well as a temporal gap detection task in which the gap was between two brief tone pulses [27,31,52,53,65]. 

The test of speech understanding in the presence of competing speech was chosen for two main reasons. Primarily, it involves a comparison of speech understanding with and without spatial separation between the target talker and two masking talkers. The calculated difference between the “colocated” and “separated” conditions is called spatial release from masking (SRM). SRM is expected to depend on binaural sensitivity and thus should be related to the ability to perform the dichotic FM task. In addition, performance on both component tasks and the derived SRM have known relationships to aging and pure-tone detection thresholds [24,66,67]. 

Data collection on the test battery, conducted with PART implemented on a portable tablet computer, took less than an hour for subjects to complete, satisfying the requirement for “rapid” testing. Note that this time included about ten minutes for two tests of tone in noise detection not reported here due to a programming error in the design of the measures. The data sample reported here, while not particularly large (41 listeners), is sufficient to demonstrate the utility PART could have in future explorations of similar relationships between these and other suprathreshold auditory assessments, cognitive assessments, or other measures determined in the future, and speech understanding.

In addition to answering the question of what happens when one attempts to use consumer-grade electronics to measure a battery of suprathreshold tests on a sample of participants varying in age and hearing loss, this study also seeks to explore the relationships among those tests and the speech in competition tasks, which have already been shown to be related to age and hearing thresholds [67]. While it is useful to know how much loss of spatial release should be expected for a listener with a particular set of hearing thresholds, after taking age into account, this does not explain why performance is reduced. It is hoped that with the right set of additional test measures, it would be possible to say with some certainty which cues an individual listener is using and which they are not. This would open the door to a wide range of counseling and rehabilitative options that are currently quite difficult to pursue, given the uncertainty about why two people with similar audiograms often have different abilities to understand speech in complex environments.

## 2. Materials and Methods

### 2.1. Participants 

Forty-one volunteers aged 23 to 80 years (mean: 51.1 yrs; standard deviation: 16.7 yrs), participated as listeners. Pure-tone hearing thresholds were obtained by an audiologist using traditional manual testing carried out in a sound-treated test room. All participants had audiometric hearing thresholds better than 85 dB HL between 0.25 and 8 kHz. Audiograms for all 41 participants are shown in Figure 1. Pure-tone average (PTA) thresholds based on an average of thresholds at 0.25, 0.5, 1, and 2 kHz ranged from −3.12 dB HL to 41.25 dB HL (mean: 15.58 dB HL; standard deviation: 11.19 dB HL). Hearing thresholds at the two ears differed by an average of 4 to 6 dB for frequencies below 2 kHz and an average of 8 to 10 dB for frequencies above 2 kHz. Two of the participants had interaural asymmetries greater than 10 dB at more than one audiometric frequency and so the regression analyses were conducted both with and without their data. Because the results of the regression analyses when those two participants were excluded were essentially unchanged (other than a reduction in statistical power) their data are included in all analyses reported below. The full data set, including Appendix A, is available at https://github.com/gallunf/SR2022 (accessed on 24 May 2022). 

### 2.2. Stimuli and Procedures 

Procedures and stimuli were a subset of those described in Diedesch et al. [14] and Larrea-Mancera et al. [12]. As for both of those studies, the tests described here (other than audiometric procedures) were conducted on an iPad running the PART application with calibrated Sennheiser HD 280 Pro headphones. These relatively inexpensive headphones were calibrated by the experimenters prior to use with a low-cost microphone and another iPad running a Sound Level Meter app, as in Gallun et al. [10]. Participants were seated in a comfortable chair in either a quiet room or a sound booth and were instructed to take breaks whenever necessary. Our previous work [12,16] has shown that test environment and even test equipment is not a significant predictor of performance for listeners using PART on a tablet or a smartphone, across a wide range of models of device and multiple headphones. For this reason, we felt comfortable using both a sound booth and a quiet room to increase the number of participants that could be tested with the time and resources available.

As described in more detail below, each participant completed a set of eight tasks, comprised of six two-cue two-alternative forced-choice (2C-2AFC) tasks assessing their ability to detect changes in various types of auditory stimuli and two speech tasks that assesses participants ability to correctly identity speech in the presence of competing talkers. 

The 2C-2AFC task has been described in detail elsewhere [12,14,68], and involves four temporal intervals, the first and fourth of which contain the standard stimulus (unmodulated or otherwise lacking the target stimulus). These are the “cue” intervals. The second and third intervals are the “forced choice” portion, as one contains the target and one contains a standard stimulus and the listener is forced to choose which contains the target. In this case, correct answer feedback was given after every trial and the size of the target signal was adaptively adjusted based on the pattern of correct and incorrect responses.

#### 2.2.1. Temporal Fine Structure 

Three different tests of TFS sensitivity were used, each presented in a 2C-2AFC paradigm with adaptive tracking to estimate threshold: diotic FM (DioFM) and dichotic FM (DichFM) thresholds [31,52,65,69] and temporal gap (TGap) thresholds [27,31]. The FM tests used a pure tone with a frequency randomized between 460 and 550 Hz on each interval. Each interval contained a stimulus that was 400 ms in duration and was presented at a level of 75 dB SPL. The intervals were separated by 250 ms of silence. The standard stimulus was unmodulated and presented diotically (identically at the two ears). For both FM tasks, the target was a 2-Hz sinusoidal phase modulation that was either the same in both ears (DioFM) or that was out of phase in the two ears (DichFM). In the DioFM condition, the phase modulation created the percept of a change in the frequency of the tone at a rate of 2 Hz. The amount of this change (the “modulation depth”) was then adjusted until the listener could just detect that the frequency was changing. The DichFM modulation, however, created a dynamic interaural time difference cue that resulted in a percept of a sound image that started at one side of the head and moved between the two ears at a rate of 2 Hz. In DichFM, the modulation depth was reduced until the depth was found at which the listeners could just detect that the sound image was moving. In both tasks, the modulation depth was adjusted on a logarithmic scale using the adaptive algorithm used by Larrea-Mancera et al. [12]. 

In the temporal gap detection task (TGap), stimuli were presented diotically, and each of the four intervals contained a pair of 4-ms 0.5-kHz tone bursts. In the standard intervals, the two bursts occurred with no temporal gap between them, while in the target interval there was a brief period of silence (a “gap”) between when one burst ended and the other burst started. The gap started at 20 ms and was adaptively adjusted on a logarithmic scale using the methods described in Larrea-Mancera et al. [12].

#### 2.2.2. Temporal, Spectral, and Spectrotemporal Modulation Sensitivity 

The target stimuli in the temporal modulation task (TM) were 400-ms bursts of broadband noise that were amplitude-modulated at a rate of 4 Hz and that the listener was asked to discriminate from the three unmodulated broadband noise standards presented in the other three intervals [70]. For the spectral modulation task (SM), the target stimulus was a spectrally modulated noise (2 cycles/octave) with a random phase that the listener was asked to distinguish from unmodulated broadband noise [58,60]. For the spectrotemporal modulation task (STM), the target noise was both temporally modulated (4 Hz) and spectrally modulated (2 cycles/octave), compared to unmodulated broadband noise. All tasks employed adaptive-staircase procedures with step sizes scaled in dB units as described in Isarangura et al. [63] but otherwise the methods were comparable to those described by Bernstein et al. [3]. 

#### 2.2.3. Speech in Competition 

Speech understanding in competition was measured using sentence-level stimuli from the closed-set Coordinate Response Measure (CRM) corpus [71] and consisted of syntactically identical, time-synchronous sentences produced by three male talkers and presented in colocated and spatially separated listening conditions [72]. Participants were instructed to attend to a target talker located directly in front of them in a virtual acoustic spatial array while ignoring two masker talkers that either were also located directly in front of the participant (colocated condition; CO) or were located at +45° and −45° (separated condition, SEP). The virtual acoustic spatial array was implemented using a generic set of head-related transfer functions (HRTFs) following the methods developed and validated by Gallun and colleagues [24,67,73]. Each target and masker sentence took the form: Ready (CALL SIGN) go to (COLOR) (NUMBER) now; the target talker always used the callsign “CHARLIE”. Each masker talker used one of seven other callsigns and different color and number combinations from those spoken by the target talker. Participants were instructed to indicate the color-number combination spoken by the target talker, using a 32-element color and number grid displayed on the iPad that contained buttons representing all possible combination of four colors and eight numbers. 

Participants were familiarized with the response matrix during a short practice session in which the target “CHARLIE” sentences were presented from directly in front at 65 dB SPL without any distractor speakers. During testing, progressive tracking was used to reduce target-to-masker ratios (TMRs) from 10 dB to −10 dB in 2-dB steps, with two trials at each target-to-masker ratio. Participants were provided with feedback about correct or incorrect responses on each trial. Following the methods developed by Gallun et al. [24], the correct number of responses out of 22 trials was subtracted from the starting target-to-masker ratio of 10 dB to approximate target-to-masker thresholds (in dB). This measure estimates the point at which performance is 50%. In addition to reporting TMR thresholds, in dB, for the colocated and separated tasks, a derived measure of spatial release from masking (SRM), in dB, was calculated by taking the difference between thresholds in the colocated (CO) and spatially separated (SEP) conditions. 

## 3. Statistical Analyses

Statistical analyses were centered on the question of the degree to which suprathreshold auditory processing abilities account for differences in individual performance that age and hearing loss cannot. Table 1 shows descriptive statistics for all predictors and outcome measures. First, correlations were calculated among the eight test measures, SRM (the difference between CO and SEP), age (operationalized as age in years at time of testing, referred to as Age), and hearing loss (operationalized as PTA). No corrections for multiple comparisons were applied, as the goal of the correlational analysis was primarily to give an idea of the strength of each variable when considered on its own. The main analysis involved backward linear regression, in which all of the predictors were entered into the model and then those not accounting for a significant proportion of the unexplained variance (at a criterion of *p* < 0.100) were eliminated. It should be noted, however, that as there were a total of 76 correlations calculated; if one wishes to consider each of the variables on its own it would be necessary to apply some form of correction for multiple comparisons. The most conservative approach would involve using the Bonferroni inequality and thus dividing the *p*-value for significance (*p* < 0.05) by 76, thus resulting in a critical value of *p* < 0.00066. All analyses were conducted in SPSS v27, which does not provide *p*-values less than 0.001, thus guaranteeing that all of the correlations would be regarded as non-significant. 

Rather than considering each correlation on its own, the approach taken here, backward regression, evaluates statistical significance based on the final model prediction. After each variable not accounting for a sufficient proportion of the unexplained variance has been removed, the final model is then evaluated for significance. The estimated effect size is based on the adjusted R^2^ value of the final model, which is a measure of the variance in the dependent variable attributable to the linear regression model prediction, and includes an adjustment for the mathematical improvement in any model when an additional predictive variable is introduced. 

To set the stage for the linear regression, the values in Table 2 are presented first for SEP and SRM, followed by correlations with Age and PTA. The significant correlations among the suprathreshold measures are then presented. None of the correlations between CO and any of the other measures reached a level of *p* < 0.05 and thus are not presented in Table 2. 

### Results: Linear Regression Modeling 

Table 3 shows the performance of the final linear regression models for each speech measure (CO, SEP, SRM) after backward elimination of all variables not significantly accounting for variance at a level of *p* < 0.100. The final regression model for CO was unable to fit a model with an adjusted R^2^ value that exceeded 0.000, and so no variables are listed. The linear regression model that provided the best prediction of thresholds in the SEP condition (adjusted R^2^ = 0.288), included Age and Diotic FM detection. For SRM, the final regression model (adjusted R^2^ = 0.310) also included Age and Diotic FM. 

## 4. Discussion 

In this analysis of 41 people of varying ages and hearing losses, with the exception of a diotic FM task, measures of suprathreshold auditory processing abilities were not strong predictors of variation in speech understanding in competing speech backgrounds. These results indicate that, while there are relationships between speech understanding and suprathreshold abilities, more work is needed to reconcile the results of the current study with the existing literature. The sections below compare these results to others in the literature and suggest a variety of potentially fruitful directions for future work in this area. Importantly, the current study demonstrates that any future investigations are likely to benefit from the further development of effective ways to test large numbers of participants on a wide range of tests.

### 4.1. The Relationships among Tests of Suprathreshold Processing and Speech Understanding

There were three speech measures examined in this study. The first, which did not include spatial separation of the talkers (colocated three talker speech; CO), showed no significant relationship with Age, PTA, or any of the other tests included in this analysis. The lack of correlation with Age and PTA is consistent with the data of Jakien and Gallun [67], who were able to explain only 5% of the variance among their listeners on the collocated condition using a model that included Age; PTA was not a significant predictor. It is possible that collecting a larger number of test runs on each participant would yield a relationship between Age and speech understanding, as stronger relationships were observed after more test runs in the Jakien and Gallun study. Nonetheless, the low proportion of variance explained for the colocated maskers by Age and PTA is consistent with the results of that study as well as others [24,56,66,74]. 

The second speech measure was the separated condition with the same three talker stimuli and task (SEP). For this condition, the Jakien and Gallun study reported that for the same amount of testing, a linear regression model based on Age and PTA explained 28% of the variance in thresholds estimated for their listeners. Here, PTA was not a significant predictor in the final regression model shown in Table 2. However, an alternative analysis included in the Appendix A did include PTA. In that analysis, the p-value to eliminate a variable was set at *p* > 0.20 rather than *p* > 0.10. In the model using the default criterion value for SPSS v27 (eliminate variables for *p* > 0.10), Age and DioFM explained 29% of the variance. This could be interpreted to mean that PTA and DioFM are tapping a similar aspect of the SEP measure, but another possibility is that the sample size tested here was simply insufficient to show an effect of PTA significant to allow to be retained in the final regression model. The two samples were very similar in age and PTA, but varied in the number of participants. Jakien and Gallun’s 82 listeners had a mean age of 46.7 years and the 41 participants tested here had a mean age of 51.1 years. The mean PTA of the participants in the Jakien and Gallun study (12.48 dB HL) was very similar to the mean of the listeners tested in this study (15.58 dB HL). However, the correlation between PTA and SEP was greater in the Jakien and Gallun study (r = 0.615) than in this study (r = 0.467), and the correlation between PTA and DioFM seen here was only 0.285 (see Appendix A). However, the correlation between Age and PTA was 0.476, which probably explains why PTA was unable to account for a significant proportion of the variance once Age was included. The inclusion of DioFM is consistent with the notion that both performance on the DioFM and the SEP tasks reflect variations in suprathreshold abilities rather than audibility, but replication with a larger sample of participants is needed to before firm conclusions can be drawn.

The third speech measure, SRM, is derived from the other two and as such, might be expected to provide little additional information. Indeed, here the correlation between SEP and SRM was the strongest observed across the entire dataset (*p* = −0.944). Nonetheless, other studies have consistently reported differences in the variables that are associated with performance in SEP and SRM [24,56,67,73,74]. For example, in the Jakien and Gallun study, PTA was a significant predictor of SRM and SEP, while Age was a significant predictor of SEP but not SRM. In that study, PTA alone accounted for 23% of the variance in SRM. In the current study, Age was a significant predictor of SRM, also accounting for 23% of the variance. As with SEP, DioFM was able to account for another 8% of additional variance in the regression model. The results of the backward regression analysis in which variables were eliminated using the *p* > 0.20 criterion included PTA in the model along with Age and DioFM. 

The role of the diotic FM detection task in predicting SEP and SRM are both consistent with the results of Strelcyk and Dau [75] who also demonstrated that TFS sensitivity was related to speech understanding in a binaural listening task. Such a relationship may be related to the need for very precise timing at the level of the cochlear nucleus for the extraction of binaural cues [76] that lead to improved speech understanding in the presence of spatially distributed sound sources. It is surprising, however, that the diotic FM task was a stronger predictor of spatial abilities than was the dichotic task, which actually involves a binaural judgment. Additional studies will be needed to better understand this unexpected result.

The poor predictive power of the STM detection task was surprising given the previous results of Bernstein and colleagues [3,4,43]. Similarly surprising is the contrasting results of Diedesch et al. [14], who reported stronger correlations with STM than were demonstrated here. One important difference between those studies and the current study is that both included a greater degree of hearing loss in their participants. Souza and her colleagues [68,77,78] have argued that listeners vary in their ability to use dynamic spectral cues to identify formant transitions in speech stimuli, and that listeners with hearing loss are more likely to have difficulty with this cue. Future work in this area would benefit from testing larger numbers of participants with an even wider range of hearing thresholds in order to capture listeners with a range of listening abilities and strategies. This will be facilitated by the increased availability of access to portable testing of the type employed in this study, thus providing easier access to large numbers of participants who are likely to vary in ways relevant to the hypotheses being tested.

### 4.2. The Importance of Cognitive Processing Abilities for Speech Understanding

Overall, the proportion of variance in the current data accounted for by even the best-fitting model was only 31%, lending support to a number of recent studies indicating that additional variables must be considered. Humes [79] and Nuesse et al. [80] both used similar statistical techniques to those reported here, but focused on cognitive variables instead of suprathreshold auditory processing, and both studies accounted for substantially more variance in the speech in noise tests they employed than is reported here. The results of Gallun and Jakien [81], who used measures of cognitive abilities to predict performance on the same speech tasks used here, are also consistent with the hypothesis that the variance left unaccounted for in the current data is likely related to differences in cognitive processing—specifically, differences in attention and working memory. In Gallun and Jakien [81], age and PTA were used as predictors along with performance on various auditory, visual, and auditory/visual working memory and attention tasks. In that case, 60% of the variance in the SEP condition was accounted for with a model that was based on PTA and auditory/visual working memory alone. In the CO condition, age and a measure of visual working memory span under conditions of uncertainty predicted 38% of the variance; and 45% of the variance in SRM was predicted by a model that included those same two variables (age and working memory span under response uncertainty). 

The relationship between cognitive abilities and auditory processing performance among older adults is not yet fully understood. Loughrey et al. [82] used a meta-analysis of 36 studies and over 20,000 patients to show that age-related hearing loss is a significant predictor of a range of types of cognitive decline. However, even among older adults with normal hearing, cognitive performance predicts poor speech understanding; Marsja et al. [9] used structural equation modeling on data from 399 older listeners (199 with and 200 without hearing loss) which indicated that cognitive performance was a strong predictor of speech understanding. 

Another important possibility is that the suprathreshold measures used here were not the ones that explain most of the variance, or that the measurement technique was too brief to provide sufficiently reliable threshold estimates. Establishing the relationship between cognitive abilities, age, and these specific auditory perceptual abilities will be an important target for future studies. As with hearing loss, the availability of portable testing will be helpful to researchers interested in testing the large numbers of heterogeneous participants necessary to test these hypotheses and will allow thresholds to be estimated with in a manner that allows the tradeoff between efficiency and accuracy to be controlled to a much greater extent than has been common with clinical research in the past. In addition to the measures described here, PART provides a robust signal processing environment that allows the researcher access to nearly every class of psychoacoustical tests that have been attempted over the past century. Furthermore, PART is constantly being upgraded and recently has been expanded to include several validated tests of cognitive function, such as working memory, divided and selective attention, response inhibition, and fluid intelligence. Multiple investigations are already underway and many more can and have been envisioned that will leverage the affordability and accessibility of PART to generate rich datasets.

## 5. Conclusions

The current study was designed to assess the extent to which suprathreshold auditory tests account for variation in speech understanding in competing speech backgrounds using a portable testing application—PART. The main result was that FM detection was the only suprathreshold ability that appeared to be strongly related to the ability to understand target speech in the presence of competing speech in which the talkers are repeating similar low-context closed-set sentences. Furthermore, this relationship was not present for speech in which there was not spatial separation between the target speech and the two competing talkers. Hypothesized relationships with detection of binaural differences and detection of spectrotemporal modulation were not observed in this data set, despite evidence in previous work supporting these hypotheses.

Future work carried out using an application such as PART will afford data collection on substantially larger numbers of participants. In addition, portable testing will allow greater examination of the relationships between speech understanding and a broad range of cognitive tests, not to mention additional assessments of suprathreshold measures, such as TFS processing and tone-in-noise perception. These studies will also address the possible influence of different types of hearing dysfunction that would be expressed as distinct non-linear relationships among the measures. The work presented here stands both as useful information about suprathreshold predictors of speech understanding and as an example of what is possible in future research using portable testing platforms.

## Figures and Tables

**Figure 1 brainsci-12-00695-f001:**
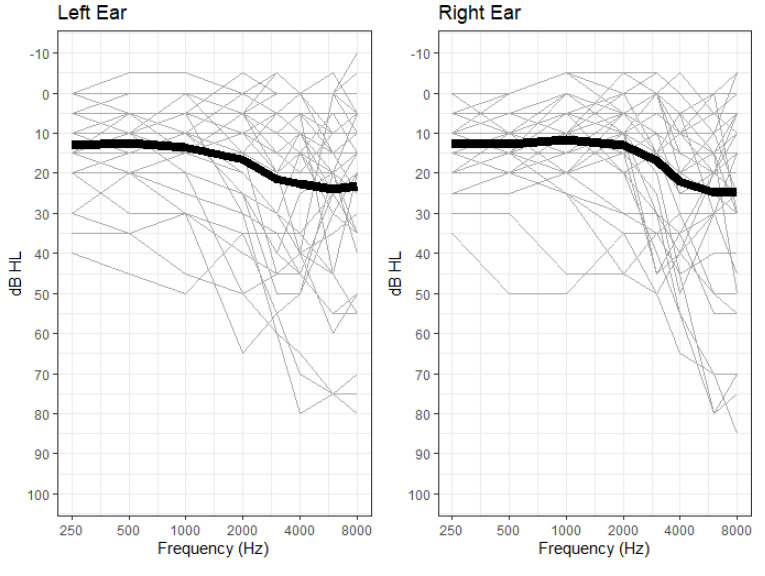
Audiograms for all 41 participants. Thin lines indicate individual listeners. Thick lines indicate mean values.

**Table 1 brainsci-12-00695-t001:** Descriptive Statistics.

	Units	Minimum	Maximum	Mean	Std. Deviation
CO	dB	0.00	4.50	2.23	1.21
SEP	dB	−9.00	5.55	−2.39	3.64
SRM	dB	−4.15	12.40	4.62	3.62
Age	Years	23.00	80.00	51.05	16.70
PTA	dB HL	−3.13	41.25	15.58	11.19
TGap	log2 (ms)	1.60	4.12	2.96	0.66
DioFM	log2 (Hz)	−2.17	3.29	0.68	1.43
DichFM	log2 (Hz)	−1.89	3.96	1.48	1.37
TM	dB	0.20	4.37	1.85	1.00
SM	dB	0.70	5.97	1.83	1.11
STM	dB	0.20	5.67	1.46	1.28

Note: dB = decibels; HL = hearing level; ms = milliseconds; Hz = Hertz; CO = speech with colocated target and maskers; SEP = speech with spatially separated target and maskers; SRM = difference between CO and SEP; PTA = 4 frequency pure-tone average; TGap = temporal gap; DioFM = Diotic FM; DichFM = Dichotic FM; TM = Temporal Modulation; SM = Spectral Modulation; STM = Spectrotemporal Modulation.

**Table 2 brainsci-12-00695-t002:** Correlations with *p*-values less than 0.05. Additional correlations available in the Appendix A.

Variable 1	Variable 2	Pearson Correlation	Sig. (2-Tailed)
SEP	SRM	−0.944	<0.001
SEP	Age	0.471	0.002
SEP	PTA	0.467	0.002
SEP	DioFM	0.381	0.014
SEP	SM	0.318	0.043
SEP	DichFM	0.315	0.045
SRM	Age	−0.497	0.001
SRM	PTA	−0.476	0.002
SRM	DioFM	−0.377	0.015
SRM	SM	−0.358	0.022
Age	DichFM	0.500	0.001
Age	PTA	0.476	0.002
PTA	TGap	0.418	0.007
TGap	DioFM	0.592	<0.001
TGap	STM	0.451	0.003
TGap	SM	0.432	0.005
SM	STM	0.691	<0.001

Note: SEP = speech with spatially separated target and maskers; SRM = difference between CO and SEP; PTA = 4 frequency pure-tone average; DioFM = Diotic FM; DichFM = Dichotic FM; TGap = temporal gap; SM = Spectral Modulation; STM = Spectrotemporal Modulation.

**Table 3 brainsci-12-00695-t003:** Final linear regression models predicting CO, SEP, and SRM.

Condition	Predictors	Adjusted R^2^	*p*	Error (dB)
CO	-	-	-	-
SEP	Age, DioFM	0.288	0.022	3.07
SRM	Age, DioFM	0.310	0.023	3.00

## Data Availability

The statistical analyses with accompanying figures for visualization are available at https://github.com/gallunf/SR2022 (accessed on 24 May 2022).

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
