# Peer review of "Relating Suprathreshold Auditory Processing Abilities to Speech Understanding in Competition"

_brainsci, 2022, doi:10.3390/brainsci12060695_

Round 1

Reviewer 1 Report

Summary

This research report investigates the speech comprehension in the presence of competing talkers and seeks to determine whether variability in “speech in competition” (hereafter referred to as SiC, though I do not suggest that the authors use this acronym) performance can be explained by individual’s performance in non-speech auditory tasks. These tasks were implemented as part of a Portable Automated Rapid Testing (PART) platform that uses an iPad and enables both lab and home-based testing.

According to the introduction, the goal of this study was to demonstrate the feasibility and usefulness of the PART platform for portable, rapid testing by showing how assessments obtained using the platform can explain variance in SiC (e.g., Line 56, Line 76). However, this doesn’t seem to fit with the conclusion, in which the goal of the study is described as assessing the potential value of ‘suprathreshold auditory tests’ for explaining SiC performance, and the authors highlight one of the assessments, temporal fine structure sensitivity, as being particularly important. In either case, the authors argue that the results demonstrate the efficacy of the platform and support the collection of metrics other than pure-tone thresholds in order to explain SiC performance.

Overall commentary

Given the rise in telemedicine, especially over the last few years, the development and validation of home-health tools and assessments is a topic of great importance. Testing the feasibility and efficacy of specific tools and platforms is an important component of development in this area, especially when sophisticated measurements are needed as in the cause of audiometry or cognitive assessment. The PART platform presented in this manuscript seems like a promising new tool for aiding in such efforts. Furthermore, given that assessments of peripheral function, such as pure tone thresholds, are not able to explain a great deal of variation in speech comprehension abilities, the use of a suite of different assessments is particularly appealing.

The abstract laid out a very clear study with promising results. However, I find that the manuscript is hampered by a lack of focus as to whether the primary goal of the study is to (1) demonstrate the feasibility of PART, or (2) to demonstrate the importance of specific assessments to explaining variability in SiC. In terms of organization, the manuscript would greatly benefit from picking one of these as the primary claim and focusing the structure of the text in that direction. Given that the authors themselves state that “this set of tests is far from comprehensive and is presented here primarily as an example of the data collection and analysis made possible by a platform such as PART,” I would suggest the more modest goal of demonstrating the feasibility of the PART platform in its ability to collect metrics that can explain any variance in this task.

In addition to improving the focus of the manuscript with regards to goals and specific hypotheses, I also think that the results/analysis section requires much more detail and motivation with regards to the methods employed, how those methods produced the results, and how the results support the authors’ conclusions. Another general comment is that in many areas, it was unclear what motivated the inclusion of specific tasks (e.g., why ‘frequency selectivity’ would be important for SiC; line 139).

This clarity is crucial, because in its current form, I do not believe that the results provide evidence for claim (2), and only limited evidence for claim (1). On Line 342, the authors themselves state (with resepct to the SRM_co results): “Note that the tests of statistical significance of the change in variance accounted for do not support the use of these multifactor models for this data set. For this reason, these results should be considered preliminary and in need of extension and replication.” Given the p-values for the models and the lack of corrections for the number of correlated predictors examined, this seems to be either a null result or weak evidence for the ability of the PART platform to explain SiC performance. I am strongly in favor of publishing null results so long as the science is sound, but they should be clearly presented as such with appropriate conclusions.

I include here a list of detailed comments and suggestions with reference to specific parts of the manuscript.

Detailed Comments

  • Line 46: Saying that specific predictors ‘drive’ performance is claiming causality between the predictors and performance. Would say ‘are associated with’ (and in general, recommend toning down strong ‘causal’ implications).
  • Line 51: The creation of accessible, portable, economical technologies is an area of critical importance and this paragraph could be expanded to emphasize this, potentially by providing some relevant statistics (e.g., average wait times for assessments, numbers of audiologists in rural areas, how far people have to travel, etc.).
  • Line 56: “Although the data set considered here does not have the sample size needed to provide sufficient power given the number of factors examined” – sufficient power to what? Determine which factors are important? If that is the case, this completely undermines the conclusion that the TFS sensitivity is an important predictor of SiC.
  • Line 66: Here and in numerous other places in the manuscript, the term ‘suprathreshold’ is used. I’m not an expert in audiometry, so this seemed strange to me, aren’t all hearing tests ‘suprathreshold’? And if there is a crucial distinction between supra and subthreshold cues/assessments, what is it? Please explain.
  • Line 85: This section seems to be a list of motivations for including specific assessments in PART. Rather than focus on auditory and cognitive abilities, I think here you could potentially explain how the PART system works, what assessments it uses (or has the potential to use), and how these specific assessments relate to peripheral/central auditory processing.
  • Line 104: “models that allow for degraded temporal cues” unclear what this means. Do you mean specific models of the auditory system, or the results of specific experiments? And are they similar to STM because they can predict performance, or because there are similarities in the structures of the models?
  • Line 116: While working memory and attention are aspects of cognition, saying that these show how ‘cognition can be an important predictor’ sounds like you are equating cognition to these two things. In general, rather than say ‘cognition’, suggest that you say ‘cognitive abilities/capacities’ or something similar.
  • Line 116: A general comment, it is unclear why discussion of cognitive abilities is relevant to this specific study, since the assessments you employ only measure perceptual sensitivity to auditory contrasts.
  • Line 121: Here you talk about how hearing loss predicts cognitive decline, which seems to run counter to the goal of the section, which is to show that cognitive abilities can affect listening performance. Would amend, explain, or remove.
  • Line 125: Similarly, instead of “cognitive performance” would say “performance in cognitive tasks” or “performance in tasks that measure aspects of cognition” or be even more specific, like ‘working memory capacity’.
  • Line 126: “Establishing the relationship between cognitive abilities and these specific auditory perceptual abilities will be an important target for future studies.” This comment would be more appropriate in the discussion.
  • Line 130: This defines a very different study goal than that presented in the conclusion.
  • Line 135: Please define diotic and dichotic for readers unfamiliar with these terms.
  • Line 139: Why is frequency selectivity important?
  • Line 144: How were the audiometric hearing thresholds obtained? Was this in a lab using specific software, or at home using the PART platform?
  • Line 145: How are audiograms converted into a single pure-tone-threshold metric? Also, seems like this sentence is missing a ‘were’ or some other verb.
  • Line 158: It’s excellent that you include the raw data and scripts in an easily obtainable format. If you could provide plain text versions of the analysis scripts for those who don’t have access to SPSS, that would be even better. Also, you should say somewhere what software you used to perform the analyses.
  • Line 163: Why were the audiometric procedures not done with PART? Is this not possible?
  • Line 165: More detail on this calibration is needed. Did the participants do it themselves? Would this kind of calibration be needed if PART were scaled up even further? This seems important for its development as a clinical tool.
  • Line 167: Why were some in a quiet room and others in a sound booth? This also seems important for telemedicine. Did the results differ at all depending on where the participants were located?
  • Line 170: “assessing their ability to detect various types of auditory 170 stimuli” do you mean ‘detect a difference between’ or ‘detect the presence of’?
  • Section 2.2.1 – 2.24: I’m curious what the others think about how the tasks used might be affected by the platform or environment? Would participants need access to a very quiet room in order for the PART tasks to be useful/accurate?
  • Line 208: What was the motivation for using this very specific task? Why not use multitalker babble?
  • Line 231: To me, it seems strange to refer to SRM_co or SRM_sep as “SRM”, because spatial release from masking is a metric derived from these two tasks (neither of which in themselves constitutes SRM). Maybe just describe the assessments as ‘colocated speech comprehension’ and ‘spatially separated speech comprehension’? Later on line 416, and in Table 3, you refer to these as CoTMR and SepTMR, which is much clearer (but you should define TMR when the metrics are introduced).
  • Line 231: Also, why is it important to measure both co-located, spatially separated, and SRM? Has it been shown that these all are important independent predictors of general hearing ability?
  • Line 232: Suggest that the table be formatted into three columns, rather than one: Assessment Type; Assessment; Description.
  • Line 236: I’m not part of the audiometry community, but the necessity of this table, particularly the skewness and kurtosis, wasn’t clear. For me, histograms and pairplots for all the metrics (and maybe Age separately) would be more useful, so that the correlations between all the task metrics can be easily visualized.
  • Line 244: How does this hypothesis relate to the overall goal of the study?
  • Line 251: I’m not familiar with this method of analysis. Please provide a citation or more in-depth motivation regarding why you chose this technique, or why you did not simply enter all variables into the regression model at the same time. While it’s potentially useful to show how much additional variance a predictor provides the model, I’m concerned that rather than showing that something is in fact a useful metric this method may simply be showing that a better model can be fit to this particular dataset but may not generalize to new data.
  • Line 253: The word ‘factors’ could be misleading, since there is no PCA or Factor Analysis being performed. Suggest ‘predictor’ or ‘independent variable’ or ‘feature’. I was actually surprised that 1) the correlation analysis was performed on each assessment + each SiC, rather than between each assessment type (to show how related the metrics are to each other, which is highly likely), and 2) nothing like PCA or Factor Analysis was used to derive latent variables that capture the shared variance across the different assessments. Have you considered such analyses?
  • Line 267: This seems to come out of nowhere. What is the motivation for this correlation analysis with respect to the overall goals of the study? I suggest adding a statistical analysis section to the methods, rather than putting everything in the results, and also adding some information in the introduction regarding how you will use the JG2018 model to support the overall goals of this study. For example, in the regression models in section 3.3., did these use the same intercepts and coefficients as in section 3.2?
  • Line 267: I was surprised that given how correlated some of the
  • Line 299: TMR is undefined.
  • Line 305: Please explain how the significance of specific predictors was assessed. For example, while the authors state that linear regression is employed, F-statistics are reported which most likely indicated that an ANOVA was derived from the linear model. Was this a Type I ANOVA?
  • Line 311: I understand why you might not want to include PTA as a predictor, because you want to see what you could predict without having to obtain an audiogram. But why not include Age as a predictor, especially when it seems to explain so much variance on its own? To me, the important test is whether there is any additional explanatory power in the PART assessments above and beyond what you could guess from Age.
  • Line 319: “If age and PTA were simply proxies for abilities such as STM and TFS sensitivity…” This seems like a specific hypothesis about the metrics utilized in this study, but Line 130 suggests that this dataset can’t test this kind of hypothesis. Also, what is the motivation for not using PTA? Is it that PTA can only be obtained in a lab setting, whereas STM and TFS can be obtained at home using PART?
  • Line 342: The fact that this is a null result should be emphasized.
  • Line 362: “These results support the exploration of variables beyond age and pure-tone thresholds alone to identify the mediators or moderators of speech understanding.” Given the null result, this needs greater explanation. The results seem to indicate that the PART assessments do not provide any additional information beyond what can be predicted from Age and PTA. I would say that you didn’t find any clear results, but there may be trends that merit further investigation.
  • Line 365: This is the first mention of the economics of the headphones. Why is this important?
  • Line 367: Using the phrase ‘strong role’ seems misleading. There was a moderate correlation between DioFM and SepTMR and SRM.
  • Line 369: Does cochlear dysfunction mean hearing loss? And how does FM sensitivity relate to the specific task of speech comprehension in competition?
  • Line 372: None of the STM metrics were significant or contributed additional variance explained, so their inclusion is not warranted.
  • Line 387: This is testable by randomly subsampling data from JG2018.
  • Line 408: Again suggesting a pair-plot that shows distribution of each variable and its correlation to other variables, to clearly show deviations from normality. Furthermore, if the assumptions of the linear regresson/anova are violated (e.g., homogeneity of residuals), then this should be clearly stated and where appropriate other types of tests (e.g., non-parametric tests) should be employed.
  • Line 426: Motivation for using the closed-set corpus should be in the introduction.
  • Line 438: What would the goal be? This would be much more relevant if the main goal of the study was just to see if PART can work and obtain something relevant.
  • Line 445: Reiterating that this seems like an strong claim that isn’t wholly supported by the results.
  • Line 450: This mention of hearing ‘phenotypes’ sound novel and am not sure how it relates to the findings of the study. Also, the presence of non-linear relationships between predictors, or of relationships between predictors at all, doesn’t seem to have been assessed or discussed.

Author Response

Please see the section in the attachment referencing Reviewer 1

Reviewer 2 Report

Brief summary 

This manuscript presents a straight-forward study investigating auditory processing tests administered through the PART application and their relationships with speech understanding in competition. The results show diotic FM and spectral modulation sensitivity having the strongest relationship with speech recognition in competition when age and PTA were included in the model. Diotic FM, dichotic FM, spectral modulation sensitivity, temporal gap, tone-in-noise spectral gap, and tone-in-noise no gap were factors in the best-fitting model when PTA and age were not included. The findings of the study contribute to our understanding of how portable testing methodologies can assess auditory perception and is a timely contribution to the literature.

General concept comments

Generally, the methods are sound and results are clearly stated. The authors did a thorough job of describing the psychometric test battery and I appreciate how all tables and figures supplement the written content. I particularly found Table 1 to be helpful in the description of the test battery. I also appreciate the transparency of the data and analyses with the inclusion of the github link. The majority of my comments relate to improving clarity of the presentation and are outlined below.

Specific comments 

Abstract, line 21: Insert “an” between “on” and “IPad”

Paragraph starting at line 74: Objectives could be made clearer if the study hypotheses were stated after the description of the relationships tested.

Line 90/111: On the first couple readings of the manuscript, it was unclear to the reader how TBI was relevant to the current study.  It would be helpful to add some clarifying text to more fully connect TBI and auditory dysfunction to the current study.

Line 110-111.  The sentence starting “Similarly, TFS processing..” would benefit from rephrasing to improve clarity. The current phrasing is awkward.

Line 42, Section 2.1: Based on the experimental question, I was expecting to find audiological speech test results, (e.g., word recognition scores, QuickSIN scores) in the dataset. If they are available it would be relevant to include given the study question.  

Line 167.  Gallun (2018) does not appear to be in the reference list.

Line 267, Section 3.2: Comparison with previous model predictions is an important evaluation, but there is no mention of this comparison prior to the description in the results. Some mention of the comparison earlier in the manuscript could alert the reader to the evaluation earlier.

Line 324: There is a word missing after ‘also’.

Figure 3: Label ‘C’ is missing from Panel C.

Line 364: Since the portability and accessibility of data collection is a compelling and timely aspect of this study, it would be helpful to identify and address this aspect earlier in the manuscript to orient the reader to the importance of study findings as they relate to the portability component.

Author Response

Please see the section in the attachment referencing Reviewer 2

Round 2

Reviewer 1 Report

The authors have substantially improved the paper. I still believe that there are some potential issues with the statistical analyses and conclusions, but after addressing these the paper would be suitable for publication.

Primarily, as with my comment on the first manuscript draft, I believe that there needs to be more explicit motivation for the statistical analysis used, which was a bivariate analysis (to determine the order in which to enter predictors) followed by forward variable selection. I’m not a statistician and had never seen this type of two-stage procedure, so I started searching to see what the literature says. One paper specifically said “don’t do bivariate selection” (Sun et al., 1996) and others said “stepwise selection is inherently problematic” (Harrel, 2017; Smith, 2018). For example, Harrel (2017) says

Stepwise variable selection has been a very popular technique for many years, but if this procedure had just been proposed as a statistical method, it would most likely be rejected because it violates every principle of statistical estimation and hypothesis testing. (pg. 67)

Harrel (2017) further recommends that if stepwise methods are to be used, backwards (“step-down”) methods perform better than forward variable selection. Using the datasets provided, I tried a backwards-variable selection procedure (using R’s built-in ‘step’ function), which selected in the following models:

CO: No variables

SEP: Age + PTA + DioFM

SRM: Age + PTA + DioFM

The p-values for Age and DioFM differ in these models (Age = ~0.038, DioFm = ~0.056). However, the maximum variance explained is 33%, fitting with the author’s own conclusions on Line 410.

It seems to me that there needs to be an explicit citation and/or explanation for why the authors used correlations + forward variable selection. Furthermore, since it does not affect the overall conclusion (that there remains a great deal of variance to be explained by non-psychoacoustic factors), I would suggest also including the results of the models fit using backwards selection.

Finally, in the Discussion, the authors interpret their finding of approximately 30% variance explained as indicative that other cognitive factors need to be accounted for in order to explain performance on Speech in Competition tasks. While I find their overall argument persuasive, it is still possible that there are other explanations (that the auditory tests did not measure all potentially relevant psychoacoustic properties, that the tests may not have been sensitive enough, etc…) that deserve at least a brief mention (even if the authors then go on to say what is more likely based on the previous results).

Other comments:

Line 106: “Neurophysiological studies in animals and electrophysiological studies in humans show that aging is associated with degradation in precise temporal coding[21,51–53].

Introducing TFS distortion also disrupts speech understanding, which has been interpreted as evidence that aging leads to reduced neural synchrony and thus increased temporal jitter[54,55].” - It seems that you are saying that TFS distortion has a more deleterious effect on speech understanding in older populations (compared to younger adults I imagine), and this is evidence for degradation in precise temporal coding in older populations (is this the same as increased ‘temporal jitter’?).

Line 283: I appreciate the inclusion of a Bonferroni correction. In Table 2, the p-values are capped at p<0.001, so I suggest including a column of adjusted p-values.

Line 442: “The current study was designed to assess the extent to which suprathreshold auditory tests account for variation in speech understanding in competing speech backgrounds using a portable testing application—PART.” - Following this sentence, I think it would be useful for the reader to have a short, one sentence summary of the main finding, i.e., that the most relevant predictors were Age followed by Diotic FM though these only explained about 30% of the variance in performance.

Line 444: “Future work carried out using an application such as PART, will afford data collection on substantially larger numbers of participants on a broad range of cognitive tests and additional assessments of TFS processing and tone-in-noise perception, as well as address the possible influence of different types of hearing dysfunction that would be expressed as distinct non-linear relationships among the measures.” - This is a very long sentence and it was difficult for me to parse, suggest breaking up into smaller sentences.  

References

E. Harrell, Regression Modeling Strategies: With Applications to Linear Models, Logistic and Ordinal Regression, and Survival Analysis. Cham: Springer International Publishing, 2015. doi: 10.1007/978-3-319-19425-7.

Smith, “Step away from stepwise,” Journal of Big Data, vol. 5, no. 1, p. 32, Sep. 2018, doi: 10.1186/s40537-018-0143-6.

G.-W. Sun, T. L. Shook, and G. L. Kay, “Inappropriate use of bivariable analysis to screen risk factors for use in multivariable analysis,” Journal of Clinical Epidemiology, vol. 49, no. 8, pp. 907–916, Aug. 1996, doi: 10.1016/0895-4356(96)00025-X.

Author Response

Reviewer 1

  • We are very grateful to the reviewer for the comments on the revised manuscript. Our responses are below.

Comments and Suggestions for Authors

The authors have substantially improved the paper. I still believe that there are some potential issues with the statistical analyses and conclusions, but after addressing these the paper would be suitable for publication.

Primarily, as with my comment on the first manuscript draft, I believe that there needs to be more explicit motivation for the statistical analysis used, which was a bivariate analysis (to determine the order in which to enter predictors) followed by forward variable selection. I’m not a statistician and had never seen this type of two-stage procedure, so I started searching to see what the literature says. One paper specifically said “don’t do bivariate selection” (Sun et al., 1996) and others said “stepwise selection is inherently problematic” (Harrel, 2017; Smith, 2018). For example, Harrel (2017) says

Stepwise variable selection has been a very popular technique for many years, but if this procedure had just been proposed as a statistical method, it would most likely be rejected because it violates every principle of statistical estimation and hypothesis testing. (pg. 67)

Harrel (2017) further recommends that if stepwise methods are to be used, backwards (“step-down”) methods perform better than forward variable selection. Using the datasets provided, I tried a backwards-variable selection procedure (using R’s built-in ‘step’ function), which selected in the following models:

CO: No variables

SEP: Age + PTA + DioFM

SRM: Age + PTA + DioFM

The p-values for Age and DioFM differ in these models (Age = ~0.038, DioFm = ~0.056). However, the maximum variance explained is 33%, fitting with the author’s own conclusions on Line 410.

It seems to me that there needs to be an explicit citation and/or explanation for why the authors used correlations + forward variable selection. Furthermore, since it does not affect the overall conclusion (that there remains a great deal of variance to be explained by non-psychoacoustic factors), I would suggest also including the results of the models fit using backwards selection.

  • We are very grateful for this suggestion, and have now used backward regression. We were unable to replicate your results unless we loosened the criterion for excluding variables to p > .20 rather than the SPSS default value of p > .10. We have included and discussed both analyses, but emphasize the analysis using the default value.

Finally, in the Discussion, the authors interpret their finding of approximately 30% variance explained as indicative that other cognitive factors need to be accounted for in order to explain performance on Speech in Competition tasks. While I find their overall argument persuasive, it is still possible that there are other explanations (that the auditory tests did not measure all potentially relevant psychoacoustic properties, that the tests may not have been sensitive enough, etc…) that deserve at least a brief mention (even if the authors then go on to say what is more likely based on the previous results).

  • We appreciate this comment and have expanded the Discussion to say this, which we definitely believe.

Other comments:

Line 106: “Neurophysiological studies in animals and electrophysiological studies in humans show that aging is associated with degradation in precise temporal coding[21,51–53].

Introducing TFS distortion also disrupts speech understanding, which has been interpreted as evidence that aging leads to reduced neural synchrony and thus increased temporal jitter[54,55].” - It seems that you are saying that TFS distortion has a more deleterious effect on speech understanding in older populations (compared to younger adults I imagine), and this is evidence for degradation in precise temporal coding in older populations (is this the same as increased ‘temporal jitter’?).

  • We are agnostic on the interpretation of these data at this time, but this is what some authors have suggested. What is clearly the case is that TFS distortion disrupts speech understanding, making younger listeners with normal hearing behave like older listeners with normal hearing. The mechanism is still to be determined.

Line 283: I appreciate the inclusion of a Bonferroni correction. In Table 2, the p-values are capped at p<0.001, so I suggest including a column of adjusted p-values.

  • Unfortunately, the Bonferroni correction is a correction to the criterion, rather than to the observed p-value, so, unlike the R2 values, SPSS v27 doesn’t provide an adjusted p-value. We found some examples, but they all require that p values be calculated to more significant digits than SPSS provides. We do now note that SPSS has a limit to the p-values it reports and further emphasize how little we believe that the bivariate significance is meaningful in this context, relative to the rankings of the variables in terms of potential to explain variance in the speech measures.

Line 442: “The current study was designed to assess the extent to which suprathreshold auditory tests account for variation in speech understanding in competing speech backgrounds using a portable testing application—PART.” - Following this sentence, I think it would be useful for the reader to have a short, one sentence summary of the main finding, i.e., that the most relevant predictors were Age followed by Diotic FM though these only explained about 30% of the variance in performance.

  • We appreciate this suggestion and have added a summary of the results.

Line 444: “Future work carried out using an application such as PART, will afford data collection on substantially larger numbers of participants on a broad range of cognitive tests and additional assessments of TFS processing and tone-in-noise perception, as well as address the possible influence of different types of hearing dysfunction that would be expressed as distinct non-linear relationships among the measures.” - This is a very long sentence and it was difficult for me to parse, suggest breaking up into smaller sentences.  

  • This has been addressed. Thanks for noting it.